# Comparative Characterization and Identification of Poly-3-hydroxybutyrate Producing Bacteria with Subsequent Optimization of Polymer Yield

**DOI:** 10.3390/polym14020335

**Published:** 2022-01-15

**Authors:** Aidana Rysbek, Yerlan Ramankulov, Askar Kurmanbayev, Agnieszka Richert, Sailau Abeldenov

**Affiliations:** 1National Center for Biotechnology, 13/5 Kurgalzhynskoye Road, Nur-Sultan 010000, Kazakhstan; ramanculov@biocenter.kz (Y.R.); kurmanbayev@biocenter.kz (A.K.); abeldenov@gmail.com (S.A.); 2Department of General Biology and Genomics, L.N. Gumilyov Eurasian National University, Kazhymukan 13 St., Nur-Sultan 010000, Kazakhstan; 3School of Science and Humanities, Nazarbayev University, Qabanbay Batyr Ave 53, Nur-Sultan 010000, Kazakhstan; 4Department of Genetics Faculty of Biological and Veterinary Sciences, Nicolaus Copernicus University in Toruń, Lwowska 1, 87-100 Torun, Poland; a.richert@umk.pl

**Keywords:** P(3HB), bean broth, ultraviolet irradiation, molecular characterization, TEM, biopolymers

## Abstract

In this work, the strains *Bacillus megaterium* RAZ 3, *Azotobacter chrocococcum* Az 3, *Bacillus araybhattay* RA 5 were used as an effective producer of poly-3-hydroxybutyrate P(3HB). The purpose of the study was to isolate and obtain an effective producer of P(3HB) isolated from regional chestnut soils of northern Kazakhstan. This study demonstrates the possibility of combining the protective system of cells to physical stress as a way to optimize the synthesis of PHA by strains. Molecular identification of strains and amplification of the *phbC* gene, transmission electron microscope (TEM), extracted and dried PHB were subjected to Fourier infrared transmission spectroscopy (FTIR). The melting point of the isolated P(3HB) was determined. The optimal concentration of bean broth for the synthesis of P(3HB) for the modified type of *Bacillus megaterium* RAZ 3 was 20 g/L, at which the dry weight of cells was 25.7 g/L^−1^ and P(3HB) yield of 13.83 g/L^−1^, while the percentage yield of P(3HB) was 53.75%. The FTIR spectra of the extracted polymer showed noticeable peaks at long wavelengths. Based on a proof of concept, this study demonstrates encouraging results.

## 1. Introduction

Plastic pollution has taken on catastrophic proportions in the world. In the last 10 years alone, more products that are plastic have been produced than in the previous century. Disposable tableware, bags, packaging, bottles and various containers are the most common types of plastic garbage that we “produce” every day, which is why approximately 800 species of animals are currently at risk of extinction due to eating and poisoning by them. Factories producing plastic products emit up to 400 million tons of carbon dioxide into the atmosphere per year. Plastic garbage litters the shores and coastal zones, clog the sewage systems of cities, and create flood threats. At the same time, only 5% of the production volume is eventually recycled and reused [1].

Therefore, a new phenomenon of our millennium is the “Great Pacific Garbage Patch”. Attempts to stop the catastrophic trend have been conducted since the middle of the XX century. Even then, environmentalists were sounding the alarm about the growing “Big Garbage Patch”, which currently, according to various estimates, covers up to 1% of the Pacific Ocean. It is an unimaginable mountain of garbage of anthropogenic origin, accumulated over the centuries in the waters of the Pacific Ocean. For many years, ocean currents have brought garbage dumped into the water to the same area. Today, its cluster is an island the size of the United States. There are probably more than 100 million tons of garbage on this site.

Plastic, as is well known, does not decompose but, over time, it breaks down into small pieces the size of plankton. In the upper layers of ocean water there are six times more plastic than plankton—light plastic does not sink and floats just a few tens of centimeters from the surface of the water, thereby attracting fish. As a result, it turns out that the same waste that we throw away comes back to us on the dining table along with food or water [2].

Over the past 50 years, plastic production has increased more than 22 times, and over the past decade alone about 180 billion US dollars have been invested in production facilities. Meanwhile, the global COVID-19 pandemic has caused a surge in the use of plastic medical masks, gloves and protective glasses, while many restrictive measures against single-use plastic products have been lifted [3].

More than 5 million tons solid household waste is generated in Kazakhstan, of which plastic makes up 10–15%. This means that every resident of the country, including an infant, uses and throws away about 42 kg of plastic waste every month. In addition, this figure is growing every year. In this regard, experts of the United Nations Environment Programme (UNEP) are actively putting forward proposals for a consolidated international action plan to address the problem of plastic waste, and the number of signatories, including plastic manufacturers, financial institutions and governments, has increased by 25% and there are now almost 500 parties, who are the main drivers of the growth of consumption of biodegradable plastics, as well as demand from developing high-tech industries (medicine, cosmetology, etc.). The largest manufacturers of biodegradable plastics are in the USA—Nature Works, in Europe—BASF, Nova Mont, and in Japan—Mitsubishi Chemicals [4,5].

One of the promising bioplastics is poly-3-hydroxybutyrate (P(3HB)). These polyesters include repeating hydroxy acyl monomers of the general formula: [–O–CH(R)–CH_2_–CO–]_n_, where R=CH_3_, and are of commercial importance due to their biodegradability and thermoplastic properties [6]. Poly(3-hydroxybutyrate) s have similar characteristics in a number of physio-chemical properties with synthetic polymers (polypropylene, polyethylene) that are widely used and produced in huge quantities that do not break down in the natural environment. In addition to thermoplasticity, polyhydroxyalkanoates have optical activity, antioxidant properties, and piezoelectric effect and, most importantly, they are characterized by biodegradability and biocompatibility. Polyhydroxybutyrate, called the biomaterial of the future, is a natural and environmentally friendly polymer derived from more than 300 different types of microorganisms. Noticeable amounts of P(3HB) accumulate in bacteria from birth: *Alcaligenes*, *Chromatium*, *Hyphomicrobium*, *Methylobacterium*, *Nocardia*, *Pseudomonas*, *Rhizobium*, *Spirillum*, *Streptomyces*, *Vibrio* and others. However, only a few microorganisms are promising industrial producers: *Bacillus*, *Alcaligenes*, *Ralstonia*, *Azotobacter* and *Methylobacterium*. Bacteria of these genera are able to accumulate polyesters on relatively inexpensive substrates (acetate, glycerin, molasses, methanol, glucose, sucrose and ethanol) with constant restrictions of certain minerals such as nitrogen, phosphorus and oxygen [7].

For many bacteria, after accumulation in the cytoplasm, the polymer serves as a source of carbon and energy during starvation. Similarly, it serves as an endogenous source of carbon and energy during sporulation. For members of the Azotobacteriaceae, it functions as an electron sink, regulating local oxygen concentrations, and acting as a carbon stock [8].

Poly(3-hydroxybutyrate) (P(3HB)) is a biodegradable polyester that can be naturally produced by microorganisms capable of converting and storing carbon in the form of intracellular granules [9]. It is known that P(3HB) is a protector for bacterial cells under various adverse environmental conditions, such as drought, thermal and osmotic shock, UV irradiation, and oxidizing agents [10]. P(3HB) accumulates in prokaryotic cells under conditions of unbalanced growth and performs the function of a reserve substance for storing carbon and energy, like fat, glycogen and starch in animals and plants. These biopolymers have a number of specific properties, such as the ability to biodegrade and compatibility with the living tissue of the body, which opens up great opportunities for their use in practice [11].

The combination of properties characteristic of P(3HB) makes it promising for use in various fields: medicine, pharmacology, food and the cosmetics industry, agriculture, utilities and radio electronics. In medicine, P(3HB) is used as absorbable suture threads, bandages, tampons, and rods in orthopedics. It is also possible to obtain flexible films, semi-permeable membranes, various hollow shapes (bottles, containers, boxes, etc.), gels and adhesives and as a polymer matrix for long-release medicines [12,13,14,15]. The potential possibility of synthesis by bacteria of polymers of various composition gives grounds for the targeted production of materials with specified properties.

The aim of our research was to isolate microorganisms from the regional chestnut soils of northern Kazakhstan (Nur-Sultan) and obtain effective bacteria that would produce P(3HB). Bacteria strains were isolated on selective media by the method of marginal dilutions.

## 2. Materials and Methods

### 2.1. Morphological and Biochemical Characterization of P(3HB) Positive Bacterial Isolates

To describe the cultural and biochemical properties, tests were applied according to the “Bergi Bacterial Determinant” [16]. The following tests were used: the ability to use various organic compounds as a source of carbon and energy (glucose, lactose, fructose, mannose, raffinose, sucrose, arabinose, maltose, trehalose), the formation of indole, hydrogen sulfide, ammonia, dilute gelatin, oxygen ratio, temperature and NaCl concentration, oxidase and catalase test, and determination of optimal pH and temperature for crop growth. The ability to reduce nitrates to nitrites, urease activity, starch hydrolysis, oxidase and catalase activity, etc. were determined.

Morphological features of preparations of stained fixed cells of isolates were studied microscopically using a trinocular microscope with transmitted light (Carl Zeiss, AxioScope A1, Oberkochen, Germany) with a total magnification of ×1000. The generic identity of the selected bacteria was established based on morphological, cultural and chemotaxonomic features [17]. Accumulation of P(3HB) in cells was noted by staining Sudan Black B. The purity of the obtained P(3HB) was confirmed by instrumental methods and its physicochemical properties were studied.

### 2.2. Molecular Identification of Selected P(3HB) Producing Bacterial Isolates

The molecular identification of the selected P(3HB)-positive bacterial isolates was carried out by analyzing them for 16S rRNA amplification. To do this, the genomic DNA of the selected isolates was extracted using the Thermo Scientific, Waltham, MA, USA. Gene JET #0721 genomic DNA purification kit. The quality and quantity of extracted genomic DNA was analyzed by running it on an agarose gel; then it was used to amplify the 16S rRNA gene using universal bacterial primers, 8F (5′ AGA GTT TGA TCC TGG CTC AG 3′) and 806R (5′ GGA CTA CVS GGG TAT CTA AT 3′). The reaction mixture consisting of a genomic DNA template, a ready-made base mixture and two primers which were amplified in a reaction cycle including one denaturation cycle at 98 °C for 1 min; 35 cycles, denaturation at 98 °C for 15 s, annealing at 55 °C for 30 s and elongation at 72 °C for 40 s; followed by a final single elongation at 72 °C for 5 min. The size of the amplified fragments was checked by electrophoresis on 1% agarose gel. The amplified and purified products of the 16S rRNA gene were specially sequenced in two directions using forward and reverse primers, and the resulting chromatograms were analyzed using the Bio Edit sequence alignment editor [18]. The final aligned consensus sequences were analyzed using the online software Basic Local Alignment Search Tool of nucleotide (blast) (http://www.ncbi.nlm.nih.gov/nucleotide. Accessed 01 November 2021) for their comparison with existing sequences in the NCBI database and their final identification [19,20]. The identified sequences were submitted to the NCBI Gen Bank to obtain a registration number for each isolate.

In addition, the strains were identified using a time-of-flight Auto flex speed mass spectrometer with matrix-activated laser desorption/ionization (Bruker Daltonik GmbH, Bremen, Germany).

### 2.3. Molecular Characterization of Selected Bacteria Producing P(3HB) by Amplification of the phbCGene

For the molecular characterization of selected bacteria producing P(3HB), a PCR protocol was developed for the amplification of the *phbC* and *phbB* genes responsible for encoding the enzyme P(3HB)-synthase, an important enzyme involved in the biosynthetic pathway P(3HB). The *phbC* and *phbB* genes of the selected 5 isolates were amplified using six different sets of primers specific to the P(3HB) synthase gene. Three sets of primers were developed using primer 3.0 software using the *phbC Bacillus* sp. gene sequences, as well as the *phbB* gene for *Bacillus* sp. available in the Gen Bank database. However, in order to obtain the amplification of the complete *phbC* and *phbB* gene, three more sets of primers were developed by generating the reverse complement of the gene sequences available in the Gen Bank database, and then selecting the 5′-terminal sequences of the original and reverse complement sequences as primers (Table 1).

A 50µLPCR reaction was carried out, including initial heating at 98 °C for 3 min, 35 cycles, subsequent denaturation at 98 °C for 10 s, annealing at 62 °C for 30 s, elongation at 72 °C for 30 s; followed by final elongation at 72 °C for 7 min. To analyze the amplification results, PCR products were subjected to electrophoresis on 1% agarose gel together with the DNA molecular weight marker O’Range Ruler DNA Ladder Mix No. 1173. The results were visualized in a transilluminator.

### 2.4. Screening of P(3HB) Producing Bacteria on a Light Microscope

For microscopic studies, samples of the corresponding colonies were prepared on slides, fixed by heating and stained with 0.3% (weight/volume in 70% ethanol) with Sudan Black B solution for 20 min. Colonies were discolored by immersion of slides in xylene, and then contrasted with safranin (5% wt./vol. in sterile distilled water) for 10 s. Bacterial cells that turned black were producers of poly(3-hydroxybutyrate) (P(3HB)), while the unpainted ones were marked as negative [21]. Positively colored isolates were selected for further works.

### 2.5. Optimization of the Nutrient Medium and Cultivation Conditions

The cultivation conditions of the producers were determined by parameters such as the availability of nutrients, a source of carbon and nitrogen, and other physical parameters such as pH, temperature and incubation time for P(3HB) production. Priyanka Lathwal et al. 2018 selected the parameters of the nutrient medium according to the study [22].

No. 1. Modified mineral medium Law and Slepecky (g/L) 20 g of glucose; 1.0 g (NH_4_)_2_SO_4_; 4.35 g Na_2_HPO_4_; 1.3 g KH_2_PO_4_; 0.2 g KCl; 0.02 g MgCl_2_; 0.001 g CaCl_2_; 0.01 g FeCl_3_; 0.001 g MnCl_2_; 0.1 g Na_2_SO_4_ distilled water up to 1.0 L; PH 7.0. [23,24].

No. 2. Mineral medium with the addition of bean broth, which we call RB (g/L) 20 g of glucose; 1.0 g (NH_4_)_2_SO_4_; 4.35 g Na_2_HPO_4_; 1.3 g KH_2_PO_4_; 0.2 g KCl; 0.02 g MgCl_2_; 0.001 g CaCl_2_; 0.01 g 0.1 g Na_2_SO_4_ and 20 g bean broth (BB), distilled water up to 1.0 L; PH 7.0.

No. 3. Bean broth (g/L) 20 g sucrose; 1.0 g KH_2_PO_4_; 0.2 g MgSO_4_ and 40 g bean broth (BB), distilled water up to 1.0 L; PH 7.0.

No. 4 and No 5. Mineral medium RB with the addition of bean broth 5.0 and 35 g/L.

The ratio of the amount of carbon with: BB was: for the medium No 1—20:0; for the medium No 2—20:20; No 3—20:40; No 4—20:5.0 and No 5—20:35.

The seed material was cultivated in 150 mL Erlenmeyer conical flasks with various mineral media in a shaker at 150–180 rpm at a temperature of 30 °C, 48 h (Innova R43, New Brunswick Scientific, Edison, NJ, USA). The stopper from the flask was wrapped with a polyethylene film to reduce the oxygen supply. It is known that with a low oxygen concentration and a high carbohydrate content, bacterial cells can be in a state of reductive stress—an overflow of NAD(F)H. To reduce it, activation of metabolic pathways that consume reducing equivalents—glycogen synthesis or P(3HB)—can occur. Due to the discharge of electrons from NAD(F)H in the synthesis of P(3HB) it is regenerated over (F)+, and thus, there inhibition of the enzymes of the Krebs cycle and therefore NADP+-dependent isocitrate dehydrogenase and citrate synthase is prevented [25].

Biomass (C) was determined by the optical density of the culture suspension at λ = 600 nm with a photoelectrocolorimeter (AP-700, APEL, Saitama, Japan) relative to the medium without cells: C = A_600_/0.134 mg/mL.

### 2.6. Extraction and Quantification of P(3HB) by Chemical Means

To obtain P(3HB) films and calibrate the fluorescent method, the Lowe method was used [26]. The isolates were cultured in a nutrient broth at 30 °C for 48 h in a shaker at 160 rpm. A bacterial suspension of 1 mL (if A_600_ = 0.2) or 2 mL (if A_600_ = 0.2) was placed in micro-tubes and centrifuged for 10 min at 6000 revolutions (Centrifuge 5415 D, Eppendorf, Edison, NJ, USA). We added 5 mL of sterile water to the sediment and stirred it using ultrasonic treatment for 5 min. We took 2 mL of suspension and added 2 mL of 2 n HCl, then heated it in a water bath for 2 h at 100 °C, after which it was centrifuged for 10 min at 6000 revolutions. The precipitate was dried, 5 mL of chloroform was added, and the tubes were tightly sealed and shaken overnight at 150 rpm and 28 °C. Then the tubes were centrifuged for 20 min at 6 thousand revolutions and extracted with 0.1 mL of chloroform. The lower transparent chloroform-containing fraction was taken into a glass vial and dried in air. A white precipitate (P(3HB)) was formed at the bottom of the vial, to which 1 mL of concentrated H_2_SO_4_ was added and heated at 100 °C for 20 min. Because of hydrolysis and dehydration, croton acid, a swamp-colored liquid, was formed. The sample was measured on a spectrophotometer.

To measure the optical density, the samples were hydrolyzed with 5 n NaOH and sterile water, bringing the pH to 4 and passed through a filter with 0.45 µ pores. The optical density of the resulting transparent solution was measured on a spectrophotometer (GE Gene Quant 100, Little Chalfont, UK) at 235 nm (the maximum optical density of crotonic acid due to the double bond), using neutralized sulfuric acid as a control, and the samples were compared with P(3HB) from Sigma Aldrich, Darmstadt, Germany. The amount of P(3HB) was calculated by the formula: [P(3HB)] (mg) = 0.0056 × A_235_ [27].

### 2.7. Obtaining an Effective Producer under Ultraviolet Irradiation

As mentioned above, P(3HB) is a protective system for the cell in various stressful situations. In this work, ultraviolet irradiation of bacteria was used as a stress factor. It is known that P(3HB) is a protector for bacterial cells under various adverse environmental conditions, such as drought, thermal and osmotic shock, UV irradiation, and oxidizing agents (generating DNA base lesions repaired by DNA glycosylases). Bacterial strains were cultured in a modified mineral medium of Law and Slepecky No. 1 and in a Burke medium for nitrogen-fixing bacteria. UV rays (wavelength 200–400 nm) were used as stress agents, the source of which was a bactericidal lamp (VIO-2,Fimet, St. Petersburg, Russia).

*Bacillus megaterium* and *Azotobacter chrocococcum* strains were grown on nitrogen-limited mineral media. Cells in the active growth phase (biomass 3–4 g/L, (if A_600_< 0.2)) were separated from the culture liquid by centrifugation, washed twice with 0.1 M MgSO_4_ solution (J. Miller, 1976) and suspended in 0.1 M MgSO_4_. The suspension of the cells was irradiated with UV from a distance of 10 cm; the irradiation time was 10 min. The irradiated and control suspensions were then used to prepare dilutions and seeding cells.

A number of standard dilutions were prepared from the treated and control suspensions and sown on Petri dishes. The cups were incubated in a thermostat at a temperature of 30 °C for 48 h. After that, the number of colonies grown was counted. The cups where the survival rate of irradiated cells was less than 1% were analyzed. To screen the accumulation of P(3HB) by bacteria, smears were stained with Sudan B dye and photos of stained cells from a microscope were compared with control samples.

### 2.8. Analysis of Selected P(3HB) Producing Bacterial Isolates in Transmission Electron Microscope (TEM)

TEM-analysis of the selected PHB-producing irradiated bacteria was carried out by preparing and drying the sample. The culture suspension was fixed by incubation in 2 mL of 2.5% glutaraldehyde solution (pH 7.2), followed by post-fixation in 2.5 mL with 1% osmium tetraoxide (OS_2_O_4_) for 1.5 h at room temperature and the suspension was washed in a phosphate buffer overnight. Then the fixed tissue was dehydrated with alcohols in ascending order of 50%, 70%, 96% and 100%. Impregnation of the suspension with epoxy resin and propylene oxide was undertaken in a ratio of 1:1 and 1:3. Polymerization of the sample in epoxy resin took place at temperatures of 37 °C, 45 °C and 60 °C. To visualize the internal ultrastructure of the body, the samples were divided into ultrathin slices with a thickness of 60 nm using an ultratome (RMC Power-Tom PC, Urbana, Illinois, USA). Furthermore, the finished sections were analyzed on a transmission electron microscope (JEM 1400 Plus, Beaverton, OR, USA).

### 2.9. Extraction of P(3HB) from Biomass

The production of P(3HB) was quantified using the Law and Slepecky method [28], and the amount of P(3HB) produced was calculated using a standard curve obtained using commercial, poly-β-hydroxybutyrate (Sigma-Aldrich, Darmstadt, Germany). The growth of a positive bacterial culture P(3HB) was granulated at 10,000 rpm at 4 °C for 10 min. The granules were then washed with acetone and ethanol to remove unwanted materials, suspended in an equal volume of 4% sodium hypochlorite and incubated at room temperature for 30 min. The mixture was then centrifuged at 10,000 rpm for 10 min for precipitation of lipid granules. The filler fluid was removed, and the precipitate was washed sequentially with acetone and ethanol. The granular polymer granules were dissolved in hot chloroform and filtered through Whatman No. 1 filter paper (pretreated with hot chloroform).

The percentage yield of P(3HB) was determined using the following Equation (1):(1)$$\mathrm{Yield}\;\left(\%\right)=\frac{\left(\mathrm{PHBs}\left(g\right)\right)}{\mathrm{DCW}\left(g\right)}100\%,$$

The volumetric productivity of bacterial biomass was determined by the following Equation (2):(2)$${\mathrm{Productivity}\;\mathrm{gL}}^{-1}h^{-1}=\frac{{\mathrm{DCWg}\;L}^{-1}}{\mathrm{Time}\;\left(h\right)},$$

The volumetric capacity P(3HB) was determined using the following Equation (3):(3)$${\mathrm{Productivity}\;\mathrm{gL}}^{-1}h^{-1}=\frac{{\mathrm{PHBs}\;g\;L}^{-1}}{\mathrm{Time}\;\left(h\right)},$$

### 2.10. Fourier Transformn Infrared (FTIR) Characterization of Extracted P(3HB)

The chemical nature of the extracted polymer was confirmed by infrared spectroscopy with inverse Fourier transform. To do this, an aliquot of 1 mg P(3HB) was extracted from various isolates and analyzed using an IR Fourier spectrometer (Nicolet iS 10, Thermo Fisher Scientific, Waltham, MA, USA). The spectra were recorded in the range of 400–4000cm^−1^. P(3HB) spectra from Sigma-Aldrich, USA were used as a standard for comparison.

## 3. Results and Discussion

### 3.1. Characterization of Selected Bacteria

#### 3.1.1. Morphological and Biochemical Characterization of P(3HB) Positive Bacterial Isolates

According to morphological and physiological-biochemical properties, most of the bacteria were from the genus *Bacillus*, Gram-positive and rod-shaped. Analysis of morphological and biochemical characteristics showed that Gram-positive rods were 10 bacteria and three Gram-negative bacteria, cultures were aerobic, and survived in a salt environment [29].

By the MALDI Biotyper analysis method, 7 strains out of 13 isolated isolates were identified as *Bacillus megaterium* (Ra 1, Ra 2, Ra 3, Ra 4, Ra 5, Ra 6, Ra 7), 3 strains Ra 1(1), Ra 1(2), Ra 1(3) as *Bacillus pumilus*, 2 strains Ra2(1), Ra 1(4) as *Bacillus simplex* and strain Az 3 as *Azotobacter chrocococcum*. Strains of the genus *Bacillus* were cultivated at 30 °C in Erlenmeyer flasks at 150 rpm, and 48 h in a modified mineral medium Law and Slepecky No. 1; the plugs were wrapped with a film to reduce oxygen access. Strain *Azotobacter chrocococcum* was cultivated at 28 °C in an Erlenmeyer flask at 150 revolutions per minute, for 72 h in a Beyerink medium. They were maintained under conditions of excessive carbon source content of the following composition (g/L): 20 g sucrose; 0.5 g (NH_4_)_2_SO_4_; 1.05 g K_2_HPO_4_; 0.2 g KH_2_PO_4_; 0.4 g MgSO_4_·7H_2_O; 0.05 g CaSO_4_·7H_2_O; 0.01 g FeSO_4_·7H_2_O; 0.006 g Na_2_MoO_4_·7H_2_O; 0.1 g CaCl_2_; 0.2 g NaCl; 0.5 g Na citrate; distilled water up to 1L; PH 7.0.For balanced growth and production of P(3HB), (NH_4_)_2_SO_4_ was also added to the medium, the optimal amount of (NH_4_)_2_SO_4_for the Beyerink medium was 0.5 g/L and 1 g/L for the Law and Slepecky No. 1 mineral medium whereas a lower salt concentration strongly limited the growth of bacteria, or suppressed the synthesis of P(3HB) [30].

When cultivating Bm 2 in medium No 3, abundant biomass growth was detected after 24 h by almost 55 g/L, but when staining cells with Sudan black B, the polymer was not practical or was weakly pronounced. In this case, we decided to expose the strains to physical stress, that is, to irradiate with UV.

Wild-type strains were cultured in a modified Law and Slepecky mineral medium to determine the amount of P(3HB) synthesis. Based on the results obtained from Table 2, we selected the most active strains Ra 5, Ra 1(1) and AZ 3 for further UV irradiation.

After irradiation, those cups were analyzed where the survival rate of irradiated cells was less than 1%. Colonies were of two forms: with a smooth surface, S-forms with smooth edges and R-forms with a rough surface, folded colonies Figure 1.

The S and R form cells were well stained with Sudan black B, which leads to the assumption that UV as a stress factor has a positive effect on the accumulation of P(3HB) by the cell. However, not all strains showed a positive effect in the accumulation of P(3HB) after irradiation, for example, Ra1 *Bacillus pumilus* was not stained with Sudan Black B after irradiation, perhaps suggesting that not in all producers, P(3HB) is a protector under UV irradiation; this may indicate a difference in the structure and course of the cycle of the main enzymes from acetyl-CoA involved in the biosynthesis of P(3HB) in the *Bacillus pumilus* strain. The cells of the *Bacillus megaterium* strain after irradiation visually acquired a seal and the color was more saturated compared to the wild type of cells. Also, the nitrogen-fixing strain of *Azotobacter chrocococcum* has obviously begun to synthesize the polymer more, according to Figure 2, cellular inclusions stained with Sudan Black B are clearly visible even with an increase of only 1000× in a light microscope (Micros, MC-300, Vienna, Austria).

However, for further work, we selected only three of the most active strains in terms of color saturation Sudan Black B: RA 5 and RAF 5 from *Bacillus megaterium* Bm 5, RAZ 3 from *Azotobacter chrocococcum* AZ 3 and wild-type *Bacillus pumilus* Ra 1.

#### 3.1.2. The Effect of the Concentration of Bean Broth on the Concentration of P(3HB) during Cultivation in Flasks

To optimize the culture medium, the mutated strain Raz 3 was cultured in a mineral medium with the addition of various concentrations of bean broth as a rich source of protein in Erlenmeyer conical flasks with a volume of 150 mL on a circular rocker at 150–180 rpm at a temperature of 30 °C, 48 h, and pH 7.0. Concentrations are indicated in Table 3.

The optimal concentration of bean broth for the synthesis of P(3HB) for the modified type of RAZ 3 was 20 g/L, at which the dry weight of cells was 25.7 g·L^−1^ and P(3HB) yield 13.83 g·L^–1^. Whereas a lower concentration of bean broth significantly limits the growth of bacteria, a high one (≥30 g/L) suppressed the synthesis of P(3HB), but stimulated the growth of biomass. Table 3 shows the dependence of biomass growth and the amount of P(3HB) by the RAZ 3 strain on the concentration of bean broth equal to the carbon concentration, which significantly stimulates synthesis and is optimal.

An increase in the content of P(3HB) in cells was found in the stationary phase, when the number of biomasses, that is, cells remained constant, and the growth of biomass was determined by a photoelectrocolorimeter at OD_600_ (AP-700, APEL, Saitama, Japan) [31]. Protein saturation of the mineral medium with bean broth and special cultivation conditions due to lack of oxygen rather stimulated the growth of biomass and synthesis of P(3HB) in the mutated strain RAZ 3 compared with the growth in the mineral and Burke medium, Figure 3.

#### 3.1.3. The Molecular Identification of Selected P(3HB) Positive Bacterial Isolates

The molecular identification of selected P(3HB) positive bacterial isolates was conducted by subjecting them to 16S rRNA amplification analysis Table 4.

#### 3.1.4. Molecular Characterization of Selected P(3HB) Producing Isolates

For the molecular characterization of the isolates obtained in this study, gene-specific primers for the amplification of the PHB synthase (*phbB*) gene were developed. Primers (BmRv—5′ TCAGCAACCCACTTTTGCATTAGCTTCCAG 3′ — BmFw—5′ GTGGCAATTCCTTACGTGCAAGAGTGGG 3′;AzRv—5′ TCAACCCTTTTACGTAGCGTCCTGGTGCAG 3′ — AzFw—5′ATGGATCAAGCCACCTCCTTCGCAAGTTTCTG 3′; BaRv—5′ TCACCGGCATTGCCATTACCATTACCATTTCCG 3′ — BaFw—5′ GTAACAGGCGGATCTAAAGGTATCGGGG 3′) were developed using Prime 3.0 software employing published PHB synthase gene sequences of other bacteria in the database. The fragment was approximately 800 bp, representing a partial *phbB* gene that was amplified in two selected bacterial isolates using a set of primers BmRv—BmFw and BaRv—BaFw. However, one pair of AzRv—AzFw primers led to an amplification of the product fragment approximately equal to 1704 bp, representing the complete *phbC* gene in all isolates (Figure 4). PHB synthase is an important enzyme involved in the biosynthetic pathway of PHB. The detection of the *phbC* gene responsible for encoding this enzyme in the bacterial genome can be considered as a sign of PHB production by the corresponding bacterial isolate. In earlier studies, PHB synthase genes were also amplified, identified and characterized to study the accumulation of PHB at the molecular level in various bacteria [32,33,34] using gene-specific primers.

#### 3.1.5. Transmission Electron Microscopy (TEM) Analysis of Selected Bacterial Isolates Producing P(3HB)

The morphology of the samples (images) was obtained using a transmission electron microscope (JEM 1400 Plus, Beaverton, OR, USA) at the Nazarbayev University Laboratory of Ultra-high-resolution Microscopy, Nur-Sultan. At an accelerating voltage of 120 kW. The images were obtained in the dark field mode in a different magnification range from ×1000 to ×50,000.

TEM images Figure 5a–d show four bacterial isolates with a high content of P(3HB). TEM analysis revealed the accumulation of dense P(3HB) granules in the cytosol of all isolates. It was noticed that the cytosol was filled with large granules (present either singly or in a nascent form). In most previous studies, 10–15 granules per cell were reported but in this study, all isolates, especially RAZ 3, showed the accumulation of larger P(3HB) granules per cell, and RAF 5 showed that there were more than 10 granules per cell. Thus, TEM images confirmed the results of quantitative studies and that these isolates were effective producers of P(3HB).

### 3.2. Characterization of Film

#### 3.2.1. The Melting Point of the P(3HB) Samples

During the research, the melting point of P(3HB) synthesized by *Bacillus megaterium* and *Azotobacter chrocococcum* was determined using a device for determining the melting point (Fisher-Johns, Melting Point Apparatus, Calgary, AB, Canada). The melting point of the P(3HB) samples obtained from Sigma Aldrich and from bacteria had similar melting points of 170 °C, which may indicate a high molecular weight of the isolated polymers.

#### 3.2.2. FTIR Characterization of PHAs

To confirm the chemical nature of PHA, we used the FTIR Nicolet iS 10 at the Nazarbayev University in Core Facilities, Analytical Chemistry Office of the Provost. Spectra were used by determining the presence of functional groups present in PHA (Figure 6). The peaks present in the ester, methylene, and terminal hydroxyl groups usually represent the polymer structure of PHAs [35]. It is known, that the exact location and intensity of a peak depends on the length of the polymer chain, concentration and crystallinity of PHA. The band at 1719 cm^−1^ is a C=O (carbonyl)–COO (ether) group and a series of intense peaks located at 1000–1400 cm^−1^ (stretching C–O), which corresponds to the complex ether group present in the molecular chain of highly ordered crystal structures. The protruding vertex, present at ~1719 cm^−1^, corresponds to a complex carbonyl (C=O) propagating vibration P(3HB) (Figure 6). The characteristic band at 2850–2965 cm^−1^ shows the presence of antisymmetric and symmetric stretching of –CH bonds (alkanes) in the –CH_3_ and –CH_2_ groups, which are usually present in pure P(3HB) [36]. The strong peaks at 2923–2975 cm^−1^ are due to the stretching of the methyl and methylene groups of C–H alkanes, which are usually demonstrated by PHA polymers [37]. Thus, the FTIR results showed not only close compliance with the results of the P(3HB) standard, but also a significant degree of purity in terms of the assignment of peaks for P(3HB) obtained after extraction.

## 4. Conclusions

This work reports on the effect of physical stress on the accumulation of poly(3-hydroxybutyrate) P(3HB) in the cells of soil bacteria in a rich nutrient medium with the addition of legume broth as a protein source. It is assumed that the presence of a substrate, operational management and special conditions, as well as subsequent processing, make this polymer commercially competitive compared to plastics of fossil origin.

In the course of the conducted studies, 13 active P(3HB) producing strains were isolated from the regional chestnut soils of Nur-Sultan, Kazakhstan, attributed after identification by the MALDI Biotyper method to the species of *Bacillus megaterium* (Ra 1, Ra 2, Ra 3, Ra 4, Ra 5, Ra 6, Ra 7), 3 strains of Ra 1(1), Ra 1(2), Ra 1(3) as *Bacillus pumilus*, 2 strains Ra 2(1), Ra 1(4) as *Bacillus simplex* and strains RAZ 3 and AZ 3 as *Azotobacter chrocococcum*. However, RAZ 3 appeared due to a mutation from the AZ 3 colony of *Azotobacter chroococcum*, but after 16S rRNA identification came out as 100% *Bacillus megaterium*, and we can assume that the culture may have been contaminated with *Bacillus megaterium* during replanting. A number of promising strains of RA 1, RAZ 3, RA 5, RAF 5 and AZ 3 obtained in the course of this study, according to TEM analysis, revealed the accumulation of dense P(3HB) granules in the cytosol of all isolates, in RAZ 3 cells the accumulation of P(3HB) granules is larger, and RAF 5 shows that there are more than 10 granules per cell. Thus, TEM images confirm the results of quantitative studies and confirm that these isolates are effective producers of P(3HB). The chemical nature of P(3HB) was confirmed by the method of FTIR spectroscopy, the characteristic band at 2976 cm^−1^ shows the presence of antisymmetric and symmetric stretching of −CH bonds (alkanes) in the –CH_3_ and −CH_2_ groups, which are usually present in pure P(3HB). In this work, the Law and Slepecky nutrient medium was modified using bean broth and it was possible to increase the yield of P(3HB) using UV irradiation of cells. The optimal concentration of bean broth for the synthesis of P(3HB) for the modified type of *Bacillus megaterium* RAZ 3 was 20 g/L, at which the dry weight of cells was 25.7 g. L^−1^ and P(3HB) yield of 13.83 g·L^−1^, while the percentage yield of P(3HB) was 53.75%. Whereas a lower concentration of legume decoction significantly limited the growth of bacteria, and a high one (≥30 g/L) suppressed the synthesis of P(3HB), it stimulated the growth of biomass. The present work demonstrated that the concentration of bean broth equal to the concentration of carbon significantly stimulated synthesis and is optimal for this isolate. Based on proof of concept, this study demonstrates encouraging results. A number of analyses confirm the quality of the selected substrate, and cheap raw materials are available for industrial production of PHA, and in combination with the practicality of this strategy it can be expanded to a pilot level for the analysis of technical and economic feasibility.

## Figures and Tables

**Figure 1 polymers-14-00335-f001:**
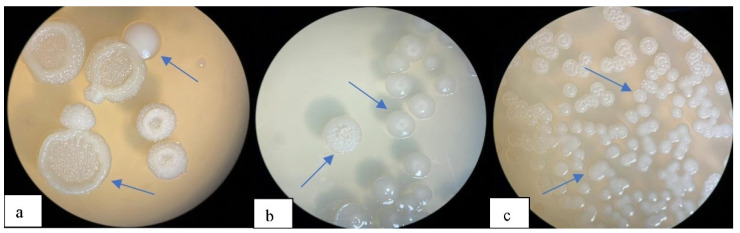
Colony after ultraviolet (UV) irradiation for 10 min. (**a**)—*Bacillus pumilus* Ra1; (**b**)—*Azotobacter chrocococcum* AZ 3 and (**c**)—*Bacillus megaterium* Bm 5.

**Figure 2 polymers-14-00335-f002:**
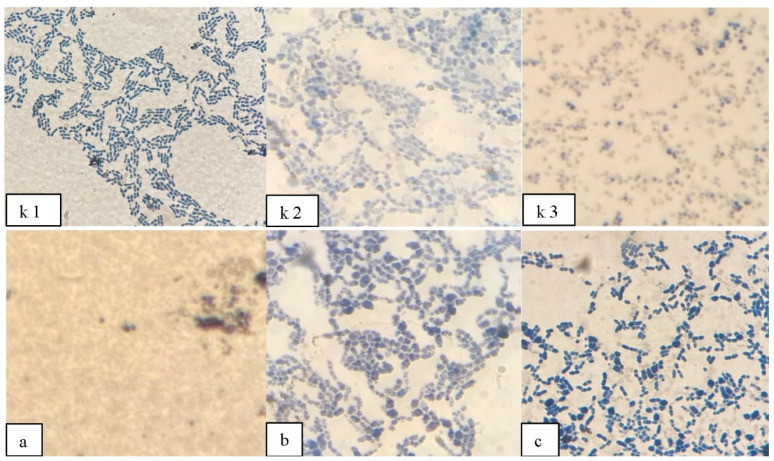
Cells stained after UV irradiation for 10 min: (**a**)—*Bacillus pumilus* Ra 1; (**b**)—*Bacillus megaterium* RA 5 and (**c**)—*Azotobacter chrocococcum* RAZ 3. Control samples, without UV irradiation: (**k 1**)—*Bacillus pumilus* Ra 1, (**k 2**)—*Bacillus megaterium* RA 5, (**k 3**)—*Azotobacter chrocococcum* RAZ 3.

**Figure 3 polymers-14-00335-f003:**
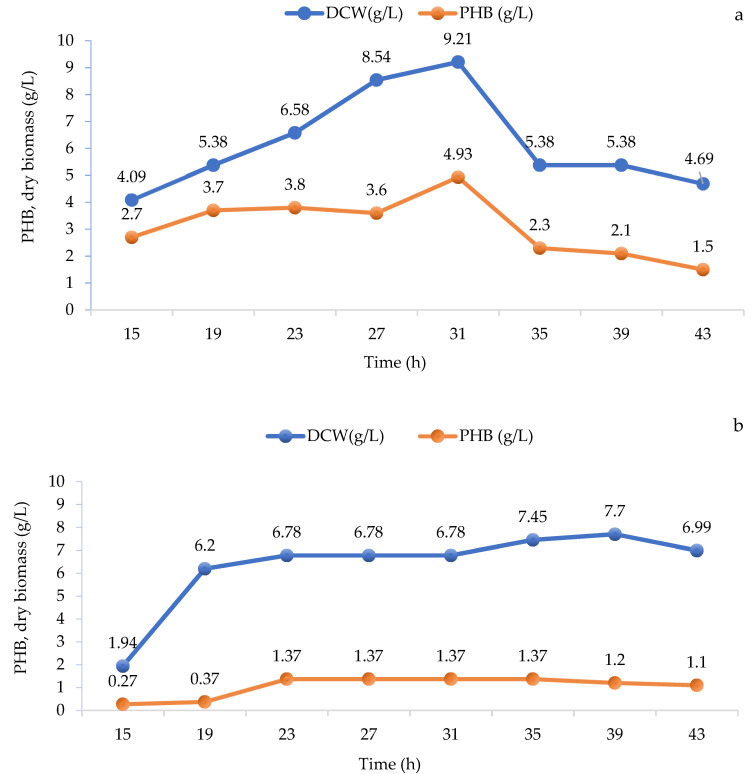
Specific rate of P(3HB) formation and specific rate of formation of residual biomass under conditions of oxygen deficiency in RB medium (mineral medium with the addition of bean broth) of strain RAZ 3 (**a**); in Burke’s medium strain RAZ 3 (**b**); in the mineral environment Ⅱ strain RAZ 3 (**c**).

**Figure 4 polymers-14-00335-f004:**
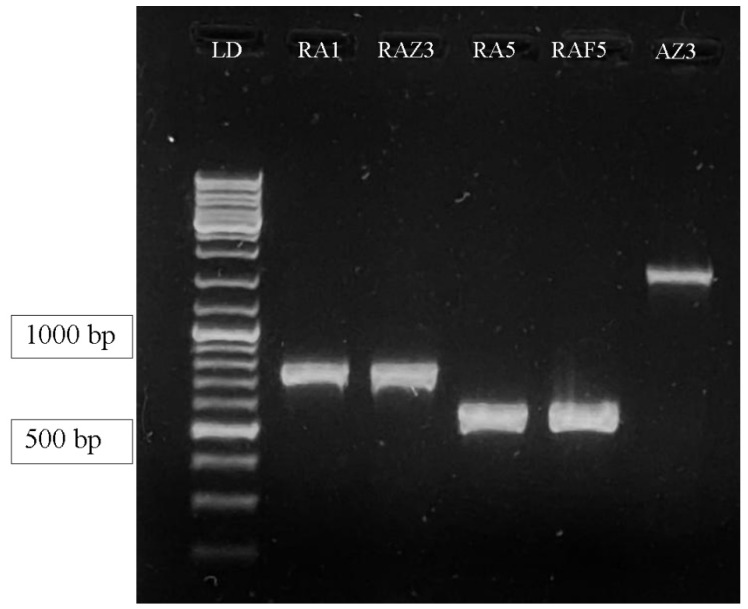
Gel showing polymerase chain reaction (PCR) amplification by primer pairs (BmRv and BmFw; BaRv and BaFw; AzRv and AzFw) specific to the *phb* gene. Track 1: DNA ladder, track 1–2 corresponding to strains RA 1 and RAZ 3 (from the left end): an amplification product of approximately 800 bp, representing the product of the partial synthase P(3HB) gene using a pair of primers BmRv and BmFw specific to the *phbB* gene. Track 3–4 corresponding to strains RA5 and RAF5: amplification product of approximately 743 bp in size is specific for the *phbB* gene using a pair of primers BaRv and BaFw. Track 5 on the right is strain AZ 3 (AzRv), an amplification product approximately 1704 bp in size, and the product is a complete P(3HB) synthase gene specific to the *phbC* gene.

**Figure 5 polymers-14-00335-f005:**
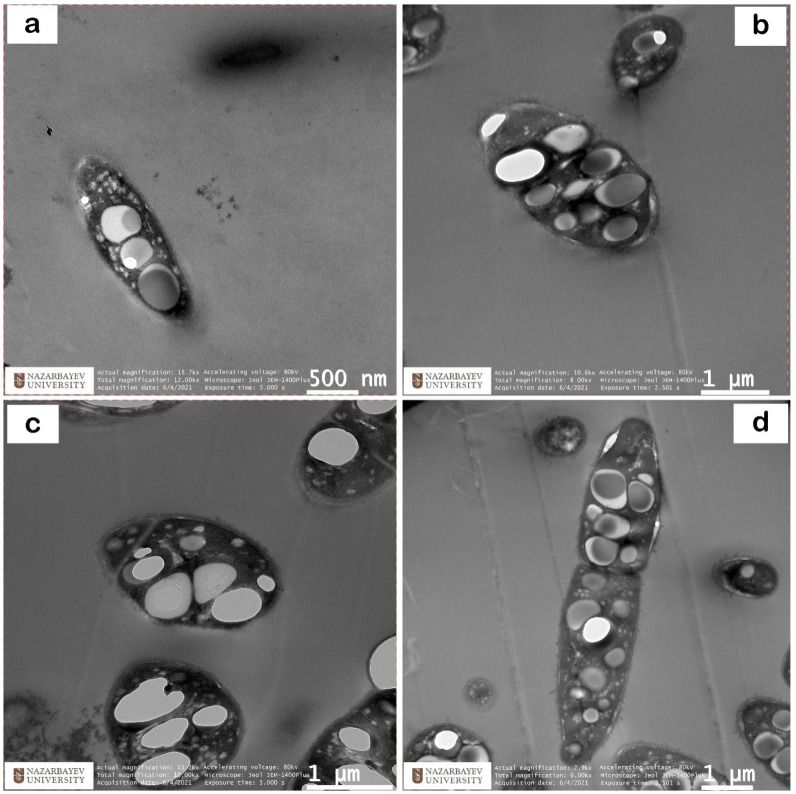
DARK images of bacterial isolates producing P(3HB) after twelve hours of incubation: half-day cultures (**a**) RA 1, (**b**) RAZ 3, (**c**) RA 5 and (**d**) RAF 5. The white granules inside the bacterial cells represent the P(3HB) granules accumulated inside them.

**Figure 6 polymers-14-00335-f006:**
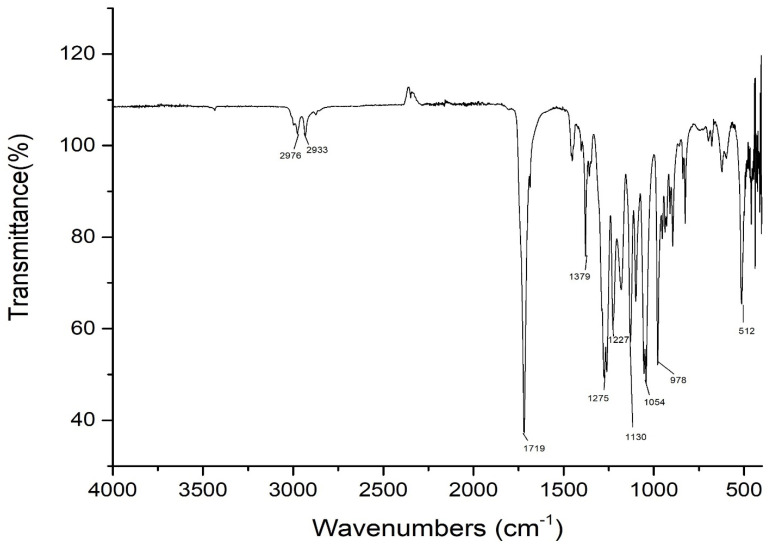
Characteristic of Fourier transform infrared (FTIR) spectra of P(3HB) obtained as a result of UV irradiation of *Azotobacter chrocococcum* AZ 3.

**Table 1 polymers-14-00335-t001:** Sequence of primers used for specific amplification of the *phbC* and *phbB* gene fragment.

P(3HB) Primers	Sequence	GC Content (%)
BaRv	5′TCACCGGCATTGCCATT ACCATTACCATTTCCG 3′	72
BaFw	5′GTAACAGGCGGATCT AAAGGTATCGGGG 3′	70
AzRv	5′ TCAACCCTTTTACGTA GCGTCCTGGTGCAG 3′	69
AzFw	5′ATGGATCAAGCCACCT CCTTCGCAAGTTTCTG 3′	74
BmRv	5′TCAGCAACCCACTTTT GCATTAGCTTCCAG 3′	71
BmFw	5′GTGGCAATTCCTTAC GTGCAAGAGTGGG 3′	72

**Table 2 polymers-14-00335-t002:** Production of P(3HB) by some soil strains in a mineral environment.

Strains	Color Sudanese Black	P(3HB) Yield (mg/mL)
Ra 1	+++ *	108.6 ± 0.88
Ra 2	+++	109.3 ± 0.87
Ra 3	++-	14.0 ± 0.17
Ra 4	+++	108.0 ± 0.89
Ra 5	+++	108.6 ± 0.88
Ra 6	+++	93.6 ± 0.72
Ra 7	++-	8.36 ± 0.57
Ra1(2)	+--	08.0 ± 0.02
Ra 1(1)	++-	108.6 ± 0.88
AZ 3	++-	138.3 ± 0.83

* +++ — saturated color; ++- — moderate color; +-- — weak color.

**Table 3 polymers-14-00335-t003:** Effect of bean broth concentration in mineral medium on synthesis of P(3HB) by RAZ3 strain.

The Content of Bean Decoction in a Mineral Environment, (g·L^−^^1^)	DCW*, (g·L^−^^1^)	P(3HB), (g·L^−^^1^)	Yield Specific P(3HB), (%)	Biomass Productivity, (g·L^−^^1^h^−^^1^)	P(3HB) Productivity, (g·L^−^^1^h^−^^1^)
0	5	0.71 ± 0.02	14.20	0.104	0.015
5.0	7.44	1.05 ± 0.07	14.11	0.155	0.022
20.0	25.73	13.83 ± 0.83	53.75	0.536	0.288
35.0	26.14	13.95 ± 0.20	53.37	0.545	0.291
40.0	28.34	14.78 ± 0.35	52.15	0.590	0.308

*DCW, (g·L^−1^)—dry cell weight.

**Table 4 polymers-14-00335-t004:** Molecular identification of selected P(3HB) positive bacterial isolates.

Strain	Strain Number http://www.ncbi. Accessed 20 April 2020	Name of the Strain	Similarity, %
AZ 3	MH763851.1	*Azotobacter chrocococcum*	99.25
RAZ 3	KT441079.1	*Bacillus megaterium*	100
Ra 5	LC430037.1 KT441079.1	*Bacillus aryabhattai*	100

## Data Availability

Data sharing not applicable.

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
