# Peer review of "Comparative Characterization and Identification of Poly-3-hydroxybutyrate Producing Bacteria with Subsequent Optimization of Polymer Yield"

_polymers, 2022, doi:10.3390/polym14020335_

Round 1

Reviewer 1 Report

Aidana Rysbek et al. report Comparative characterization and identification of poly-3-hydroxybutyrate producing bacteria with subsequent optimization of polymer yield. The work has revealed the production of poly-3-hydroxybutyrate P(3HB) by bacteria. The work is highly informative and the authors have performed several controlled experiments too. The authors have shown bacterial isolates produced P(3HB). The topic is very exciting and the study is very informative. This work is of high interest for readers working in the field of biodegradable plastic poly-3-hydroxybutyrate P(3HB) polymers. 

Thus, it requires minor revisions in order to meet the journal's requirements. 

  1. Page 7 authors have used a different font(which is not clear) for the equations. I think it´s a mistake, or the authors can explain it.
  2. Page 7; the authors have explained the MALDI biotyper analysis experiments, I wonder if the authors can present the experimental images of it or not.
  3. Page 8 table 2. The production of P(3HB) is random. A proper scientific explanation is missing. 
  4. The authors have not tried to isolate and characterize P(3HB) by GC-MS, GPC, NMR, etc. Those analytical tools would have been interesting. 

Reviewer 2 Report

From scientific point of view it is an interesting manuscript and should be published. However, i do not agree with Authors that "this work demonstrates a high potential for industrial production of P(3HB)" - because described in this paper an eperimental procedure is quite complicated and requires an application of many chemicals and many changing parameters of a biochemical process. It is also not claear which colony of bacteria is most effective for potential practical applications for syntheses P(3HB) even on a larger laboratory scale. Much more work will be necessary for commercialization of this technology of P(3HB) preparation.

1. In line 63 "tons" should be added.

2. In line 151 a phrase "electrophoresis in agarose gel on 1% agarose gel" should be corrected.

3. An abbreviation CDW in Table 3 and Fig. 3 should be exolained.

4. "g/L−1" in Table 3 shlould be changed for "g/L" or "g.L−1".
